**mSystems®**

# A Genomic Survey of the Natural Product Biosynthetic Potential of Actinomycetes Isolated from New Zealand Lichens

Peng Hou,[a,b] Vincent V. Nowak,[a,b,c] Chanel J. Taylor,[a] Mark J. Calcott,[a] Allison Knight,[d] ⓘ Jeremy G. Owen[a,b,c]

[a]School of Biological Sciences, Victoria University of Wellington, Wellington, New Zealand
[b]Centre for Biodiscovery, Victoria University of Wellington, Wellington, New Zealand
[c]Maurice Wilkins Centre for Molecular Biodiscovery, Auckland, New Zealand
[d]Department of Botany, Otago University, Dunedin, New Zealand

**ABSTRACT**   Actinomycetes are prolific producers of industrially valuable and medically important compounds. Historically, the most efficient method of obtaining compounds has been bioactivity-guided isolation and characterization of drug-like molecules from culturable soil actinomycetes. Unfortunately, this pipeline has been met with an increasing number of rediscoveries, to the point where it is no longer considered an attractive approach for drug discovery. To address this challenge and to continue finding new compounds, researchers have increasingly focused on alternative environmental niches and screening methods. Here, we report the genetic investigation of actinomycetes from an underexplored source, New Zealand lichens. In this work, we obtain draft genome sequences for 322 lichen-associated actinomycetes. We then explore this genetic resource with an emphasis on biosynthetic potential. By enumerating biosynthetic gene clusters (BGCs) in our data sets and comparing these to various reference collections, we demonstrate that actinomycetes sourced from New Zealand lichens have the genetic capacity to produce large numbers of natural products, many of which are expected to be broadly different from those identified in previous efforts predominantly based on soil samples. Our data shed light on the actinomycete assemblage in New Zealand lichens and demonstrate that lichen-sourced actinobacteria could serve as reservoirs for discovering new secondary metabolites.

**IMPORTANCE**   Lichens are home to complex and distinctive microbial cohorts that have not been extensively explored for the ability to produce novel secondary metabolites. Here, we isolate and obtain genome sequence data for 322 actinomycetes from New Zealand lichens. In doing so, we delineate at least 85 potentially undescribed species, and show that lichen associated actinomycetes have the potential to yield many new secondary metabolites, and as such, might serve as a productive starting point for drug discovery efforts.

**KEYWORDS**   biosynthetic gene cluster, genome mining, natural product, nonribosomal peptide, polyketide

The majority of antimicrobial agents currently in use are either biosynthesized directly by microorganisms or chemically synthesized based on the structures of microbial metabolites (1). Unfortunately, the introduction of novel drugs into clinical practice has almost always led to drug resistance, significantly shortening the drug's shelf life. Antibiotic resistance is estimated to cause 10 million deaths per year by 2050 if no effective action is taken (2). However, screening traditional ecological niches like soils in search of highly prolific producers of bioactive secondary metabolites frequently results in the rediscovery of microorganisms and small molecules. To address these challenges, researchers have started investigating underexplored environments in a quest to find new bacterial species capable of producing previously unknown compounds (3, 4).

Address correspondence to Jeremy G. Owen, jeremy.owen@vuw.ac.nz.

The authors declare no conflict of interest.

Lichens are self-supporting symbioses, share mutualistic partnerships between heterotrophic mycobiont (fungus) and autotrophic photobiont ("algae"), and have associated bacterial communities that benefit from and contribute to the overall fitness of the lichen holobiont (5–7). Actinomycetes isolated from lichens have been reported to produce new congeners of biomedically relevant natural products. For example, skyllamycin D, identified from the New Zealand lichen *Pseudocyphellaria dissimilis*-associated *Streptomyces anulatus* VUW1, exhibited more potent bioactivity against *Bacillus subtilis* E168 compared to previously reported skyllamycin congener (8). A *streptomyces* isolated from a British Columbia lichen *Cladonia uncialis* is the producing strain of uncialamycin. This enediyne chemical displayed broad-spectrum bioactivity (9). However, compared to the lichen dominant fungal partner, lichen dwelling actinomycetes are relatively underexplored as a source of bioactive natural products (6).

New Zealand is home to an abundance of lichen species, many of which are not found elsewhere in the world (6). Here, we use a genetics-first approach to evaluate actinomycetes from New Zealand lichens as a potential source of new bioactive compounds. A collection of 480 putative actinomycete isolates comprising 299 genetically distinct strains (≤99.5% average nucleotide identity) were obtained from 46 locations across New Zealand. Genome sequence data for each isolate was generated (Illumina HiSeq, PE150bp), resulting in a total contig length of 3 Gb, from which 8,541 BGCs were reconstructed. A majority of the resultant BGCs had high biosynthetic divergence compared to a collection of 1.2 million BGCs identified from previous sequencing efforts, shared low similarities to the functionally characterized BGCs present in the MIBiG database, and showed low relatedness to 4,309 annotated actinobacteria genomes in the NCBI Reference Sequence Database. This study demonstrates the feasibility of working with cost-efficient sequencing data and illustrates a pipeline for systematically investigating large-scale BGC data sets. By exploring the previously underestimated microbial community associated with New Zealand lichens, we show that lichen associated actinomycetes might provide a fruitful avenue for the genetics driven discovery of new secondary metabolites.

## RESULTS AND DISCUSSION

**Strain isolation, genome sequencing, and taxonomic classification.** A total of 402 lichen samples were collected from 46 locations in New Zealand (Fig. 1a). A total of 480 actinomycetes-like bacteria were recovered from 200 lichen samples. From the resulting isolates, genomic DNA was extracted, NGS libraries were prepared, and sequencing was performed for downstream analysis. Reads were trimmed and assembled, resulting in 406 assemblies, the quality of which was evaluated using CheckM (10) (Fig. S1a and b).

To assign phylogenies to the newly sequenced isolates, GTDB-Tk (11) was used for taxonomic classification at the whole-genome level. Seventy-four assemblies that fell below the number of amino acids threshold (10%) in GTDB-Tk MSA (multiple sequence alignments) step were filtered out. In the remaining 332 isolates for which genome sequence data were obtained, 10 isolates from five genera were not actinomycetes and thus were classified as off-targets and excluded from the downstream analysis. A total of 322 isolates can be classified as belonging to 16 actinobacterial genera, including *Streptomyces*, *Nocardiopsis*, *Rhodococcus*, *Gordonia*, *Nocardia*, *Microbacterium*, *Oerskovia*, *Amycolatopsis*, *Spirillospora*, *Mycobacterium*, *Streptomyces_B*, *Micromonospora*, *Rothia*, *Kribbella*, *Williamsia_A*, and *Embleya*. Only 162 (50.31%) isolates can be further classified down to the species level. Our collection recovered 39 distinct known species from six different actinobacterial genera as delineated by GTDB-Tk (11). Clustering the remaining 160 unassigned isolates was carried out using a MASH (12) distance cut-off of 96%. This clustering led to an additional 85 distinct clusters that likely represent previously undescribed actinomycete species (Fig. 1b). The current taxonomic grouping and assignment were supported by an independent pairwise genome distance estimation using Mash (12) (Fig. S2).

The 322 actinobacteria were sourced from at least 15 lichen genera, including 33 isolates from *Pseudocyphellaria*, 21 from *Collema*, 17 from *Peltigera*, and 251 from other

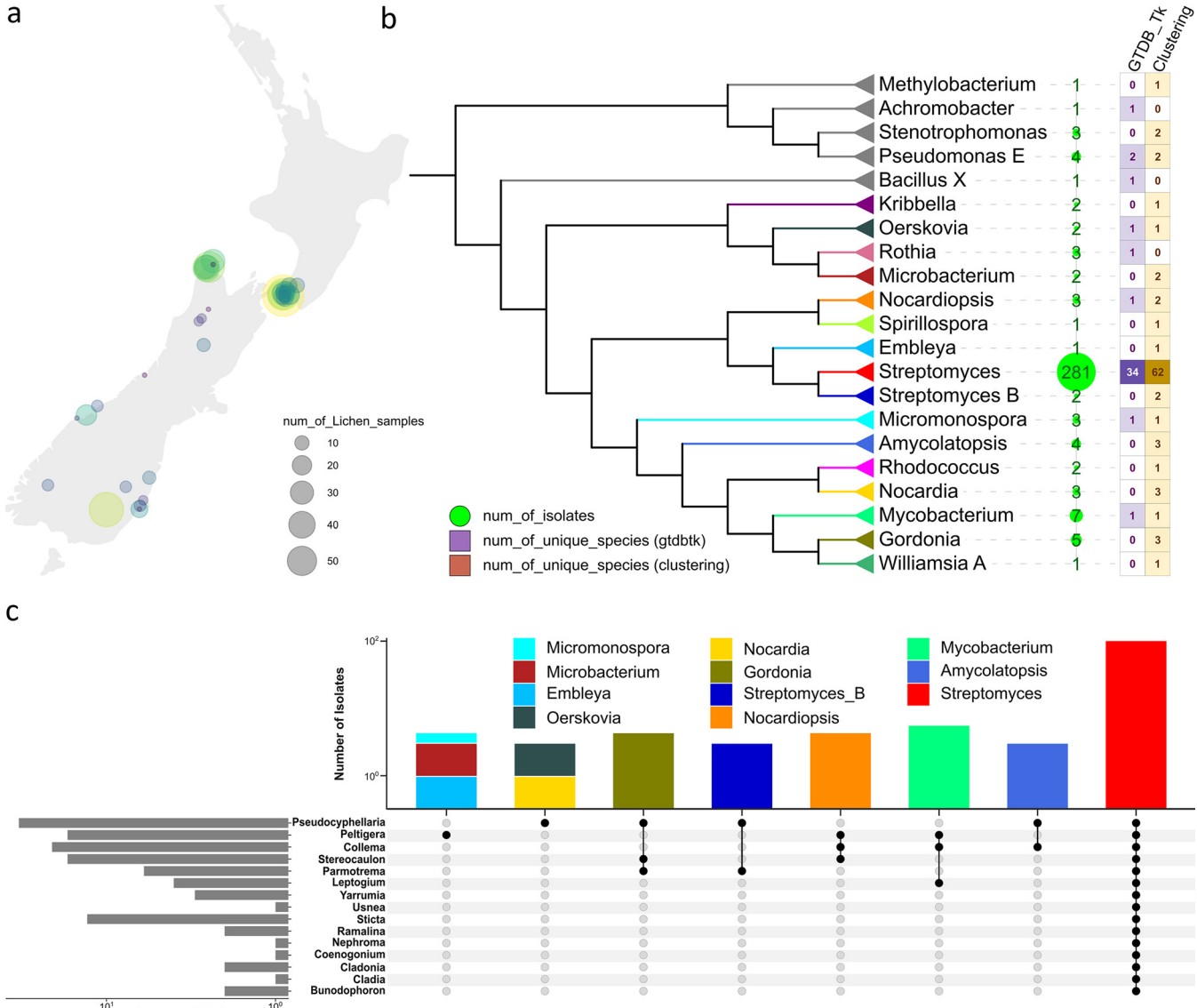

**FIG 1** Sampling information and whole-genome level taxonomic classification. (a) Sampling locations and the number of lichens sampled for this study. (b) Taxonomic cladogram of isolates collected in this study (collapsed at the genus rank). Shapeplot of isolates in log scale and heat maps of unique species recovered within each genus are provided. (c) A breakdown of the actinobacteria genera by the lichen origins (genera level) reveals "co-occurrence" patterns in certain lichen samples and actinobacteria isolates.

lichen sources or yet-identified lichen samples. While *Streptomyces* could be isolated from all lichen samples, certain actinobacteria isolates displayed a "lichen-specific pattern." For example, actinobacterial species in the genus *Embleya* were only isolated from lichens in the genus *Peltigera* (Fig. 1c). Sample details and the analysis workflow were summarized and provided in Data Set S1 and Fig. S1c.

**Identification and categorization of natural product biosynthetic gene clusters.** Actinobacteria are well-known prolific producers of antibiotics, and a majority of the secondary metabolites are encoded by BGCs (13). As a next step toward exploring the abundance of BGCs, antiSMASH 5.1.0 (14) was used to identify BGCs from our 322 draft genomes. In total, we identified 8,541 BGCs from 28,531 contigs (>5,000 bp; Fig. 2a). The average length of identified BGCs was 31.9 kb (Data Set S2). The longest BGC is 247.9 kb long from the isolate Li4c-A12.

Among the identified BGCs, we found 1,614 nonribosomal peptides (NRPs), 1,430 polyketides (PKs), 1,108 ribosomally synthesized and post-translationally modified peptides (RiPPs),

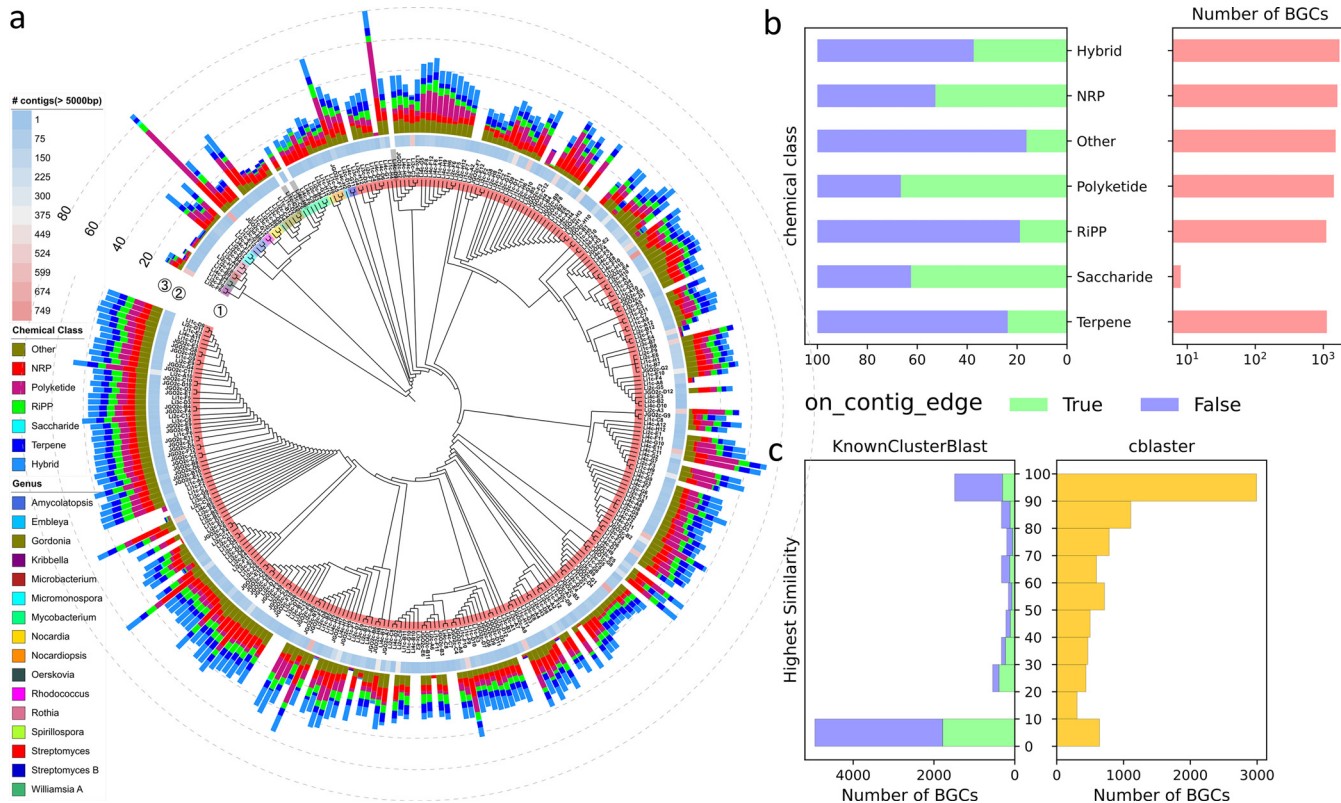

**FIG 2** Overview of the BGC diversity. (a) Taxonomic cladogram of isolates collected in this study (at the species rank). Genera boundaries ① were color coded using the same color scheme as Fig. 1b. Additional information: ② number of contigs longer than 5,000 bp and ③ chemical classes were provided. Note that some of the biosynthetic "rich" isolates were the artifact of the fragmented assembly. Assemblies that contain contigs shorter than 5,000 bp were highlighted in gray. (b) A breakdown of the portion of "complete" BGCs (on_contig_edge = False) and "incomplete" BGCs (on_contig_edge = True) by chemical classes. (c) Distplot depicts the distribution of the similarity levels of identified BGCs when queried against the MIBiG entries (left) and RefSeq records (right).

1,129 terpenes, eight saccharides. A total of 1,521 BGCs can be classified into the other chemical classes listed in Table 1. Another 1,731 BGCs have multiple cluster types assigned (Fig. 2a).

The fragmented nature of short-read assembly makes assembling long mega synthases such as NRPSs and PKSs difficult. Within the 8,541 BGCs, 3,179 BGCs (37.40%) regions were located on the contig edge; 51.86% (851/1641) of the NRP BGCs and 66.57% (952/1430) of PK BGCs in our data set were on contig edge, which suggested that they were potentially incomplete (Fig. 2b; Data Set S2). Future studies could consider using a hybrid assembly strategy (combining long-read and short-reads sequencing) to overcome the BGC reconstruction challenges. Waschulin and colleagues found that compared with using the short reads only, applying the hybrid assembly method to reconstruct the BGCs from Antarctic soil bacteria metagenome, the on_contig_edge BGC rate dropped from 96.7% to 39.8% (15).

We next compare our lichen BGCs against known, experimentally characterized BGCs in the MIBiG database using the KnownClusterBlast module embedded in antiSMASH (14, 16). A substantial proportion of "complete" BGCs (59%: 3,163/5,362) showed no significant similarity (similarity <10%) with any entries in the MIBiG database (Fig. 2c left; Data Set S2).

We also queried our 8,541 actinobacteria BGCs against 4,309 annotated actinobacteria genomes in the NCBI RefSeq Database using cblaster (17, 18). This search (Fig. 2c right) indicated that less than half of the actinobacteria biosynthetic regions derived from New Zealand lichens have close relatives among previously sequenced genomes (46.67%: 3,986/8,541, similarity >80%), and greater than 10% of BGC regions were highly distinct from the existing actinobacterial genomes in the RefSeq database (11.28%: 963/8,541, similarity ≤20%).

**TABLE 1** Distribution of antiSMASH cluster types (nonhybrid) for BGCs present in this study

| Chem_class | Chem_subclass | Count | Chem_class | Chem_subclass | Count |
|---|---|---|---|---|---|
| NRP | CDPS | 14 | Other | NAGGN | 1 |
| | NRPS | 1305 | | Arylpolyene | 12 |
| | NRPS-like | 295 | | Betalactone | 85 |
| Polyketide | PKS-like | 36 | | Blactam | 18 |
| | T1PKS | 852 | | Butyrolactone | 273 |
| | T2PKS | 158 | | Ectoine | 290 |
| | T3PKS | 351 | | Furan | 5 |
| | hglE-KS | 24 | | Fused | 4 |
| | transAT-PKS | 5 | | Hserlactone | 1 |
| | transAT-PKS-like | 4 | | Indole | 21 |
| RiPP | LAP | 37 | | Ladderane | 8 |
| | TfuA-related | 14 | | Melanin | 220 |
| | Bacteriocin | 481 | | Nucleoside | 16 |
| | Lanthipeptide | 355 | | Other | 69 |
| | Lassopeptide | 151 | | Phenazine | 12 |
| | Linaridin | 62 | | Phosphonate | 5 |
| | Thiopeptide | 8 | | Siderophore | 481 |
| Saccharide | Amglyccycl | 2 | Terpene | Terpene | 1129 |
| | Oligosaccharide | 6 | | | |

**Assignment of BGCs to gene cluster families using BiG-SLiCE.** In order to approximate the relatedness of our BGCs to a larger collection of previously sequenced BGCs, we performed a BiG-SLiCE query for our lichen BGCs against a precomputed reference model consisting of 29,955 Gene Cluster Family (GCFs) that are available as part of the BiG-SLiCE package (19, 20). These GCF models are derived from 1.2 million BGCs sourced from publicly available genomes and metagenome assembled genomes (MAGs) and were computed using a linear clustering model based on biosynthetic feature absence/presence and similarity to a reference collection of pHMMs.

We were able to assign our 8,541 BGCs to a total of 1,071 GCFs, indicating that our data set recovered a promising diversity of BGCs. For each query BGC and its matching GCF model, a distance value ($d$) was calculated. A $d$ value can be subclassified into three categories: "core" ($d \leq 900$), "putative" ($900 < d \leq 1,800$), and "orphan" ($d > 1,800$), which indicates the relationship between each BGC and its centroid GCF (19, 20). A high $d$ value ($d > 900$) implies that a BGC is a good candidate for encoding previously unknown compounds. Among our 8,541 BGCs, 5,146 (60.25%) BGCs were classified as "core" members of their GCFs. The memberships of 3,073 (36%) BGCs were "putative," implying that they were only moderately related to the centroid GCFs. A total of 322 (3.77%) BGCs were "orphans," meaning they were highly divergent from the existing 1.2 million BGCs and the GCFs to which they were assigned, and were good candidates for producing highly novel chemical entities.

BiG-SLiCE categorized our 8,541 BGCs into five chemical classes: class-Polyketide (2,193), class-other (1,807), class-NRP (1,621), class-RiPP (1,539), class-Terpene (1,340), and class-Saccharide (41). The "orphan" BGCs were relatively prevalent in class-Saccharide (26.83%: 11/41), class-Polyketide (8.39%: 184/2,193) and class-NRP (2.47%: 40/1,621) (Fig. 3a).

The "orphan" BGCs span 12 actinobacterial genera and were relatively more abundant in recently established and less sequenced genera (21) *Embleya* (11.86%: 7/59 BGCs), *Williamsia* (7.69%: 1/13 BGCs), *Amycolatopsis* (7.59%: 12/158 BGCs), and *Nocardiopsis* (6.56%: 4/61 BGCs). While *Streptomyces* genera comprised most of the isolates (281) and BGCs (7,762) in this study, the "orphan" rate in *Streptomyces* was only 3.65% (283/7762 BGCs) (Fig. 3b). When linking our identified BGCs to their lichen sources (Fig. 3c), we found that lichen genera *Pseudocyphellaria*, *Stereocaulon*, and *Peltigera* were the most prolific sources of BGCs, contributing 877, 540, and 462 BGCs, respectively. High proportions of "orphan" BGCs were recovered from the lichen genera *Leptogium* (10.17%: 12/118 BGCs),

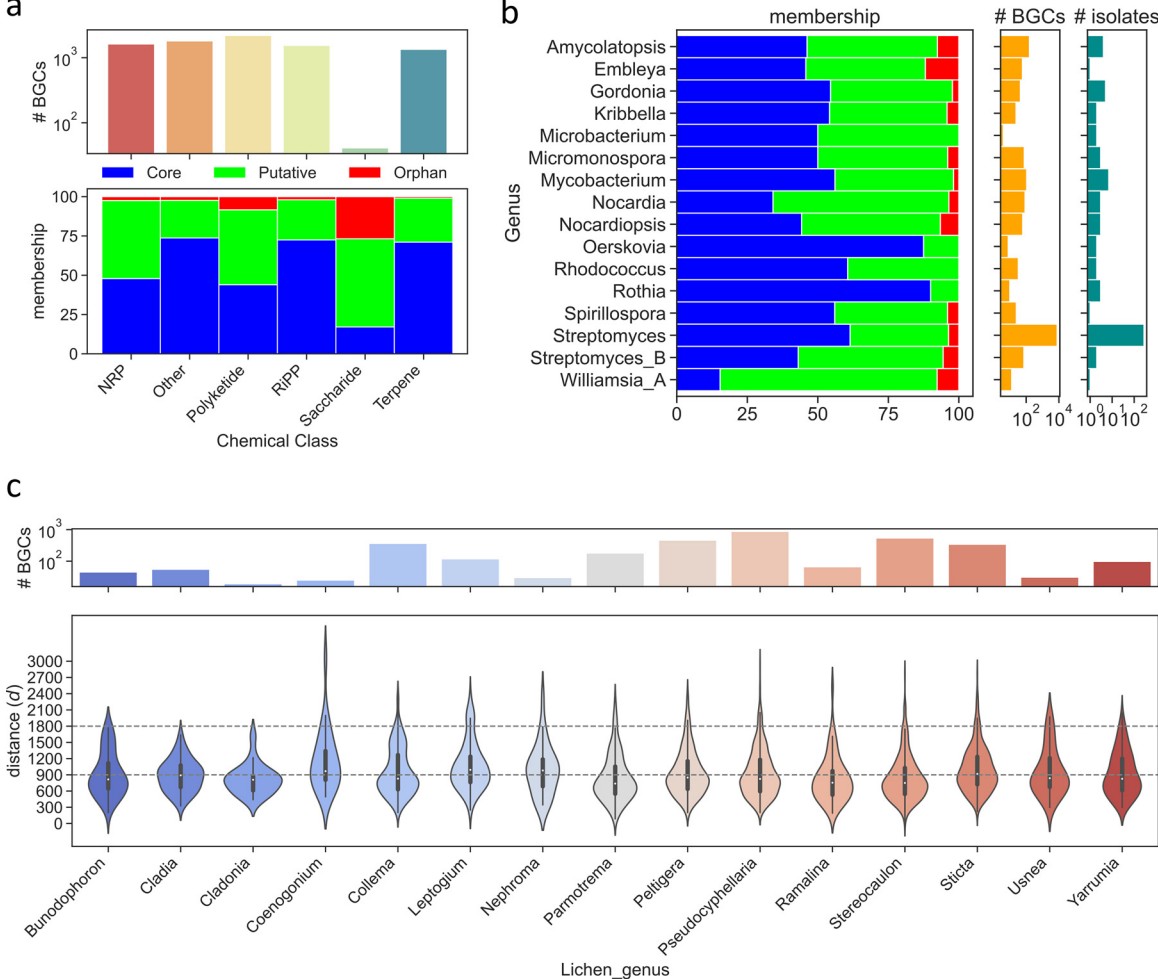

**FIG 3** BiG-SLiCE membership analysis. Distance value ($d$) was calculated for each query BGC and the GCF model assigned. The $d$ value can be subclassified into three categories: "core" ($d \leq 900$), "putative" ($900 < d \leq 1,800$), and "orphan" ($d > 1,800$). which indicates the novelty of query BGCs. (a) BiG-SLiCE chem_classes distribution of BliG-SLiCE memberships in lichen BGCs. (b) Actinobacteria taxonomic distribution of BiG-SLiCE memberships in lichen BGCs. (c) Lichen sources distribution of BiG-SLiCE memberships in lichen BGCs.

*Coenogonium* (8.00%: 2/25 BGCs), and *Usnea* (6.45%: 2/31 BGCs). This part of the analysis indicated that future studies toward gaining novel BGCs could be more effective if lichen genera are targeted.

Further manual examination of some of the identified orphan BGCs revealed that they were likely to encode previously undescribed compounds. For example, BiG-SLiCE (19) assigned one 199-kb-long Type I polyketide synthases (T1PKS) BGC Li3d-B6_r1c5, identified from an *Amycolatopsis* isolate, an orphan membership ($d = 2,051$) to GCF_200963. The GCF_200963 contains 1,250 core members and 1,427 putative/orphan members, with T1PKS (99.8%) and *Streptomyces* (54.4%) serving as the representative class and taxon, respectively (20). BGC0000073, a reference MIBiG BGC encoding macrolactone halstoctacosanolide A, is the most relevant member ($d = 549$) within GCF_200963 (16, 20, 22). Cblaster (18) search for homologous gene clusters from actinobacteria reference genomes revealed that Li3d-B6_r1c5 shows moderate homology to a region from *Amycolatopsis australiensis* DSM 44671 genome. Clinker synteny (23) and antiSMASH (14) analysis showed that Li3d-B6_r1c5 had no identical ORF organisations to these reference BGCs (Fig. 4a). The orphan membership assignment of Li3d-B6_r1c5 and different PKS domain organisations compared to the references suggested that this lichen BGC is involved in biosynthesising a novel (macrocyclic) polyketide (Fig. 4b; Data Set S3).

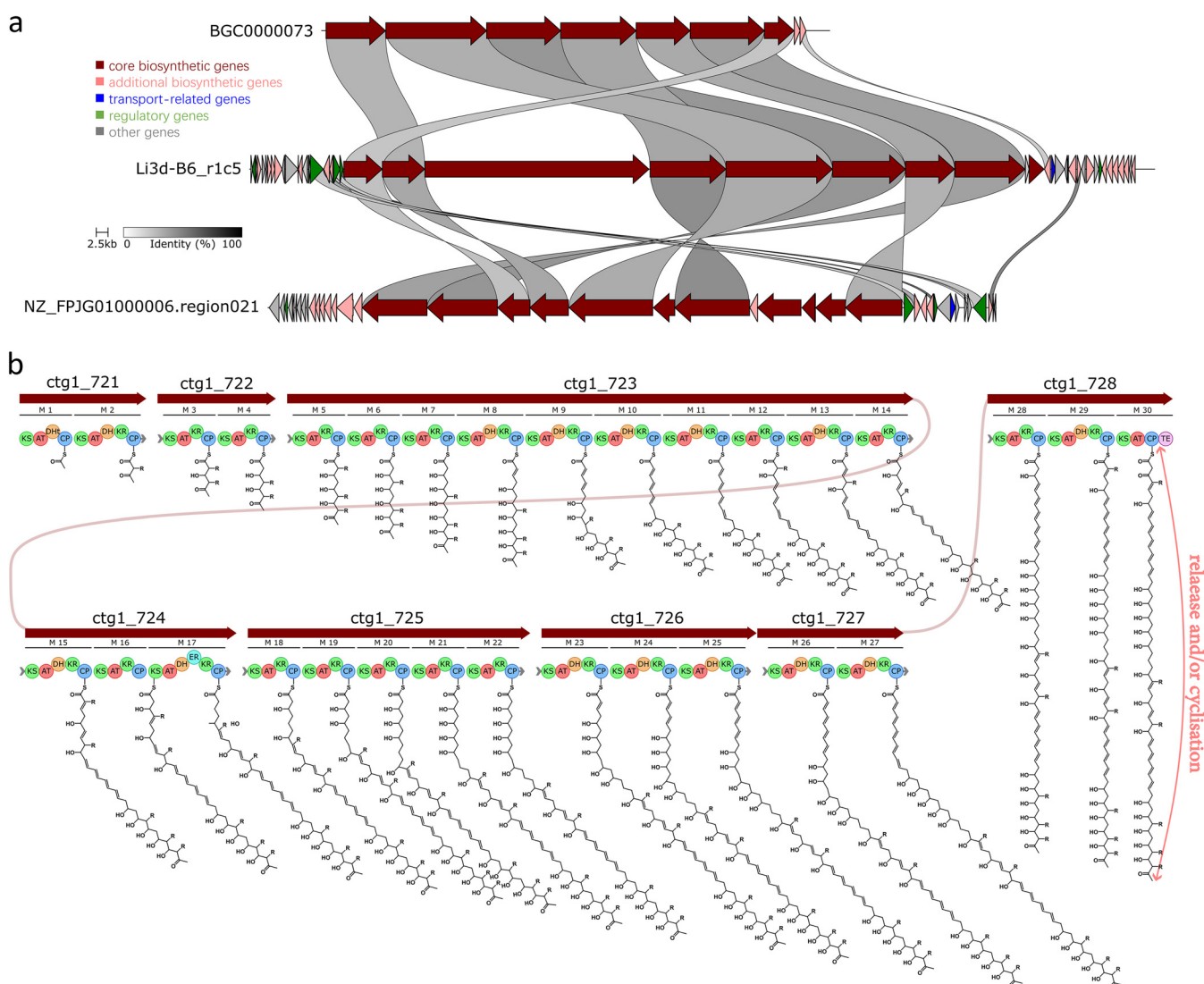

**FIG 4** BGC analysis for one orphan BGC identified from BiG-SLiCE analysis. (a) Clinker synteny analysis showed that the two reference BGCs, BGC0000073 (a T1PKS consisting of 18 modules) and NZ_FPJG01000006.region021 (a T1PKS consisting of 24 modules) and lichen orphan BGC Li3d-B6_r1c5 (a T1PKS consisted of 30 modules) have different ORF organizations. (b) Proposed core Li3d-b6_r1c5 biosynthesis. Monomers were predicted using antiSMASH NRPS/PKS monomer predictions module, and structures were drawn based on the assumed PKS collinearity. KR stereochemistry was not considered.

**Comparison to other environmental isolate data sets using BiG-SLiCE.** In order to ascertain how the diversity and novelty of our lichen data set compared to other recently sequenced collections, we compared our data (lichen) to two environmental data sets: insect-associated *Streptomyces* (24) (insect) and biofilm-forming marine microbes (25) (marine).

We grouped 4,142 insect BGCs and 274 marine BGCs into 845 and 134 GCFs, respectively (Fig. 4a). The Venn plot indicated that 42 GCFs were common among the three environmental data sets, and there was more overlap between lichen and insect, two data sets dominated by *Streptomyces* species. Among the three data sets, lichen had a higher percentage of unique GCFs (63.12%: 676/1,071) than insect (54.08%: 457/845) and marine (57.46%: 77/134) data sets (Fig. 5a). The proportion of BGCs that were classified as orphans ($d > 1,800$) varied among the data sets. In the marine data set, 2.55% (7/274) were classified as orphans, whereas just 0.82% (34/4,142) of the insect BGCs had been assigned orphan memberships. These metrics indicated that the lichen data set possesses abundant unique biosynthetic diversities (Fig. 5a) and is relatively novel

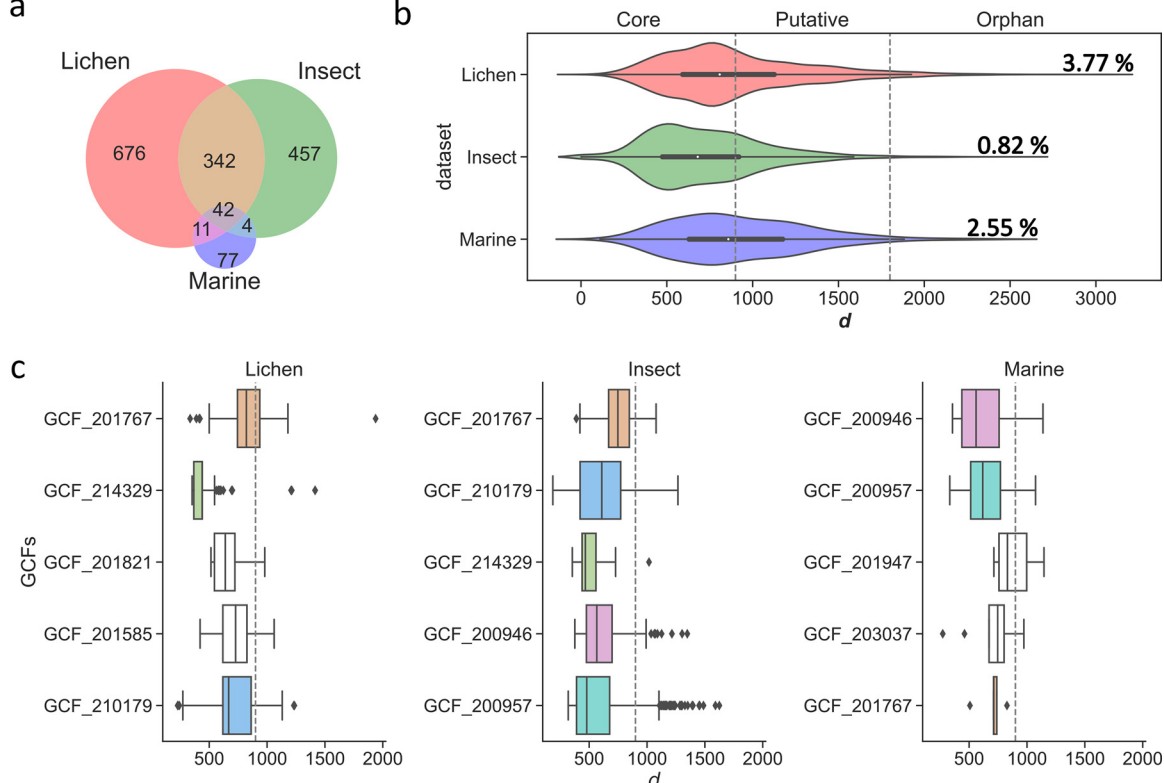

**FIG 5** Comparison with insect and marine environmental data sets. (a) Venn diagram showed the common and unique GCFs among the three environmental data sets (lichen, insect, and marine). (b) The Violin plot showed the distribution of the memberships (core: $d \leq 900$; putative: $900 < d \leq 1,800$; orphan: $d > 1,800$) for the lichen, insect, and marine data sets. (c) The boxplot of the distributions of $d$s for the top 5 prevalent GCFs (as defined by GCF shared by how many genomes) from each environment data set (lichen, insect, and marine). Note that most of the $d$s were below 900, which indicated that those BGCs were relatively conserved. The same GCFs are highlighted in the same colors. GCF_201767: Other_ectoine; GCF_214329: Other-siderophore; GCF_201821: RiPP-bacteriocin; GCF_201585: Other-siderophore; GCF_200957: NRP-generic; GCF_210179: Terpene-terpene; GCF_200946: RiPP-unknown; GCF_201947: Terpene-terpene; GCF_203037: Other-hserlactone; GCF_216237: Other-resorcinol.

(3.77% BGCs were orphans) (Fig. 5b). The percentage of orphan BGCs in our lichen data set was close to that previously reported for a metagenomics analysis of an Antarctic soil microbiome, where 3.9% BGCs from the collection of uncultivated bacteria were classified as orphans (15).

Additionally, we extracted and compared the top 5 most populated GCFs from each environment (lichen, insect, and marine). Analyzing the predominant GCFs among the three environments (to some extent) demonstrated the common and unique pathways/ strategies the microbes utilized to provide fitness benefits for dwelling in these three distinct environments (Fig. 5c). GCF_201767 was widespread among all three data sets. The representative chem_class of GCF_201767 was ectoine. Ectoines are compatible solutes and are synthesized via a biosynthetic pathway that is readily identifiable (26). The common occurrence of the ectoine gene cluster family suggested that ectoine is widely synthesized and adopted by bacteria to mitigate osmotic stress. Siderophores (GCF_214329, GCF_201585) were prevalent in the lichen and insect data sets but not in the marine collection, possibly indicating the differing abundance of iron. The majority $d$s within each prevalent GCF was below 900 among the three data sets, indicating the related BGCs were "core" memberships, and thus, relatively conserved (20).

**Molecular families-based gene cluster grouping.** Comparison to the precomputed GCF models using BiG-SLiCE gave us insights into the biosynthetic divergence and diversity of our lichen-sourced BGCs. BiG-SLiCE converted the full-length input BGCs into BiG-SLiCE feature vectors, and the GCF memberships were then assigned based on the absence/presence of particular biosynthetic features. However, it should

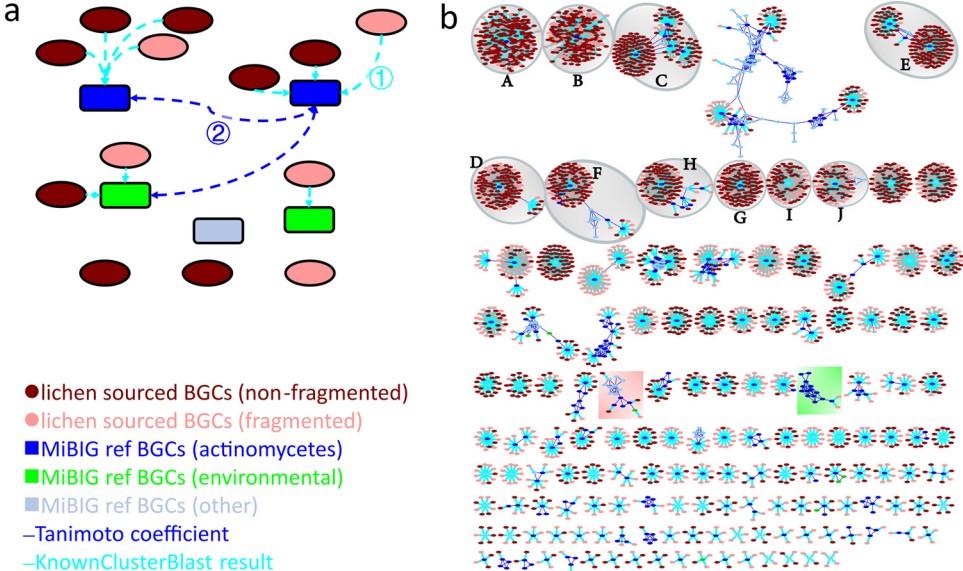

**FIG 6** Chemical space-guided gene cluster grouping. (a) Construction of Molecular Families network. Lichen sourced BGCs were first linked (1) to the MiBIG reference BGCs based on the KnownClusterBlast results. The associated MiBIG BGCs were further linked with each other based on Taniomto chemical similarity (2). (b) Molecular Families network. Unconnected lichen sourced BGCs and MFs with a count less than 6 were removed. The complete version was provided in Fig. S5a. The top 10 MFs to which lichen sourced BGCs were assigned were highlighted. A, ectoine; B, hopene; C, melanin; D, geosmin; E, desferrioxamin B; F, coelichelin; G, isorenieratene; H, SGR PTMs; I, spore pigment-related; J, streptobactin.

be noted that the hybrid state and the boundary of one BGC will significantly affect the distance ($d$) calculation and potentially lead to the overestimation of the BGC novelty (Fig. S3). Moreover, in a small-scale analysis using the MIBiG database,(16) we found that GCFs were not necessarily representative of molecular families (MFs, as defined and grouped by chemical similarities) (Fig. S4). Hence, we cannot directly infer the chemical diversity and divergence for our data sets only based on the number of GCFs and $d$s.

As an additional measure to examine potential chemical novelty, we inferred the chemical space our BGCs occupied by combining two analyses: KnownClusterBlast to assign compounds to BGCs and clustering into molecular families (MFs) using the Tanimoto coefficient (27). The MIBiG database (16) is a repository for the annotated sequences of experimentally characterized BGCs and their corresponding encoded compounds. KnownClusterBlast (28) is an analysis module embedded in antiSMASH, which compares identified BGCs with BGCs in the MIBiG repository and renders similarity scores during comparison (Fig. S3). We began by associating our 8,541 characterized BGCs (oval-shaped nodes, cyan edge) with the most similar MIBiG BGC based on the KnownClusterBlast results with a similarity cut-off of 0.2. MIBiG BGCs (rectangle-shaped nodes, blue edge) were then linked if their associated metabolites share a chemical similarity above an empirical threshold (Tanimoto similarity score $>0.5$) (Fig. 6a).

Using this chemical space-based networking approach, 3,591 lichen-sourced BGCs (42.04%, oval-shaped nodes) identified in our data sets can be directly connected (cyan edges) to 361 MIBiG BGCs and grouped into 288 MFs (Fig. 6b). Among the top 10 MFs to which our lichen-derived BGCs were assigned were ectoine (Fig. 6b, A), hopene (Fig. 6b, B), melanin (Fig. 6b, C), geosmin (Fig. 6b, D), desferrioxamin B (Fig. 6b, E), coelichelin (Fig. 6b, F), isorenieratene (Fig. 6b, G), SGR PTMs (Fig. 6b, H), spore pigment-related BGCs (Fig. 6b, J), and streptobactin (Fig. 6b, J).

We next compared the BGCs-references grouping efficiency between the BiG-SLiCE model and our MFs model. We conducted the GCF assignments for the MIBiG BGCs in the MFs and our lichen sourced BGCs simultaneously. Using the BiG-SLiCE model,

3,652 (42.76%) lichen-sourced BGCs can be co-assigned with 601 MIBiG BGCs into 170 GCFs. In our chemical-space guided network, 42.04% BGCs (3,591/8,541) can be cogrouped with 598 MIBiG BGCs, suggesting that the BGCs-references clustering efficiency of these two models are similar under the current threshold (T = 900 for BiG-SLiCE, KnownClusterBlast similarity >0.2). However, the overlap between the two sets was low (only 1,638 lichen BGCs and 305 MIBiG reference BGCs can be found in both sets), which indicated that the GCF and MF models might be complementary, each allowing grouping of different collections of BGCs. In the case of grouping lipopeptide and glycopeptide related BGCs we discussed below, we found that the chemical-guided MF model was more efficient (Fig. S6); however, this may vary on a case-by-case basis.

**Chemical space-guided congener discovery.** Genetics-driven approaches are playing an increasingly important role in searching for novel therapeutics to address the antibiotic (resistance) crisis (29–31), As such, we sought to explore our newly established network for lichen sourced BGCs for the potential to produce lipopeptide and glycopeptide antibiotics: two families of natural products that have both medical relevance and a history of successful discovery using genomics first approaches (29, 32, 33).

We began by examining the cluster of lipopeptides (Fig. 6b pink shaded area) with a focus on the actinomycetes/eDNA subcluster (Fig. 7a, the complete version was provided in Fig. S5b). This subcluster contains five $Ca^{2+}$-dependent cyclic lipodepsipeptide (CDA) MiBIG BGCs and three lichen-sourced BGCs. Through Clinker (23) synteny analysis, we noticed that one lichen BGC Li4c-G7_r1c4 shares many similarities with daptomycin (34), A54145 (35), and taromycin (36), three extensively studied CDAs (Fig. 7b). The fatty acyl-AMP ligase and the acyl carrier protein (genes in orange) were located upstream of the NRPS genes (genes in maroon). Auxiliary genes such as ABC transporters (genes in blue) and MbtH protein (genes in green) can also be found among the above BGCs (Fig. 7b; Data Set S3) (34, 35, 37). Detailed Stachelhaus analysis of the A domains (Fig. 7c; Data Set S3) revealed several conserved structural features among the above BGCs: (i) amino acids at positions 2, 9, and 10 (Fig. 7b) are conserved in d-configuration, Gly and Asp analogues, respectively; (ii) the presence of a canonical tetrapeptide motif Asp-X-Asp-Gly for $Ca^{2+}$ binding (Fig. 7b, gray shaded). Discrepancies in A domain substrate specificity and the common features shared among Li4c-G7_r1c4 and CDA reference BGCs suggested Li4c-G7_r1c4 might encode a series of compounds that are related to the reported CDA, but have different amino acid compositions in the peptide backbone.

Another molecular family of potential interest was the glycopeptide antibiotics related to vancomycin (GPAs) (Fig. 6b green shaded area; Fig. 8a; Fig. S5c). We found one lichensourced BGC Li3d-F5_r28c1 (incomplete) is 75% similar to a MiBiG reference BGC (MIBiG accession: BGC0000326) encoding the TypeV GPA isocomplestatin. Further targeted resequencing of this isolate enabled us to get the full length of Li3d-F5_r28c1 BGC. Clinker synteny (23) and antiSMASH (14) analysis showed that the ORF organisations of Li3d-F5_r28c1 and BGC0000326 were strikingly similar (Fig. 8b left; Data Set S3): several NRPS (genes in maroon) were flanked by transporter-related genes (genes in blue), halogenase (genes in pink), and cytochrome P450 (genes in green). Our BGC also contained the X-domain conserved on GPA related BGCs for cytochrome P450 recruitment (38) (Fig. 8b left). Stachelhaus alignment (Data Set S3) on the A domains revealed that our Li3d-F5_r28c1 BGC incorporates tryptophan, a characteristic moiety of Type V GPAs (39). Notably, the methyltransferase domain was absent from our lichen BGC compared to the isocomplestatin BGC (Fig. 8b left). The biosynthetic organization suggested that the lichen BGC Li3d-F5_r28c1 is involved in the biosynthesis of a new demethylated isocomplestatin-like compound (Fig. 8b right).

The above examples demonstrated that our chemical space-guided gene cluster grouping method precisely clustered the glycopeptide related BGCs (Fig. 6b green shaded area) and lipopeptide related BGCs (Fig. 6b red shaded area). However, when using two publicly available GCF models for gene cluster grouping, we found that some linkages among lipopeptide BGCs were absent, and glycopeptide BGCs were either not well-grouped or over-grouped with unrelated BGC types (Fig. S6). Chemical

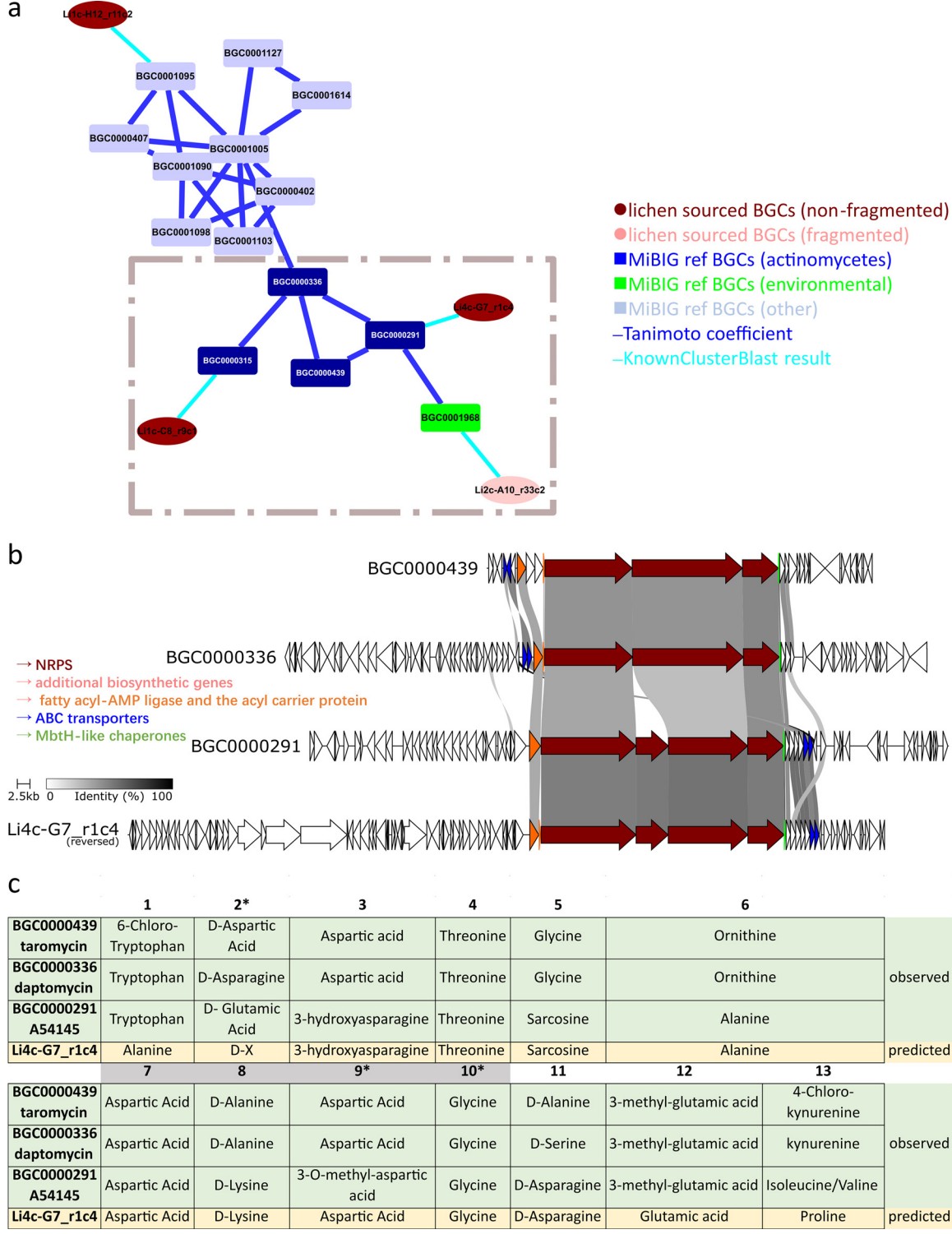

**FIG 7** CDA MF: (a) CDA MF extracted from Fig. 6b (pink shaded area). (b) Through Clinker synteny analysis, we identified one lichen BGC Li4c-G7_r1c4 that shared many similarities with reported CDA BGCs. (c) Detailed Stachelhaus alignment of the A domains for lichen BGC and reported CDA BGCs revealed conserved structural features and unconserved A domain substrates.

space-guided grouping of BGCs analysis might serve as an additional tool that is complementary to existing methods for the discovery of pharmaceutical valuable congeners. This method may be of broader utility as an additional means for clustering and categorizing BGCs to prioritize strains or cloned BGCs for further study.

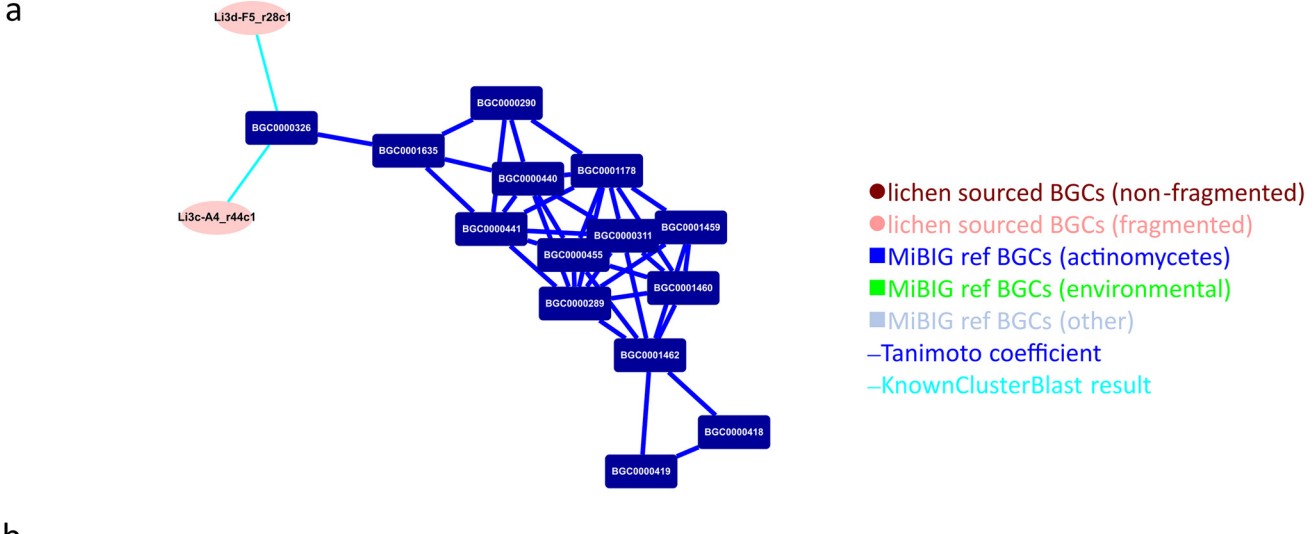

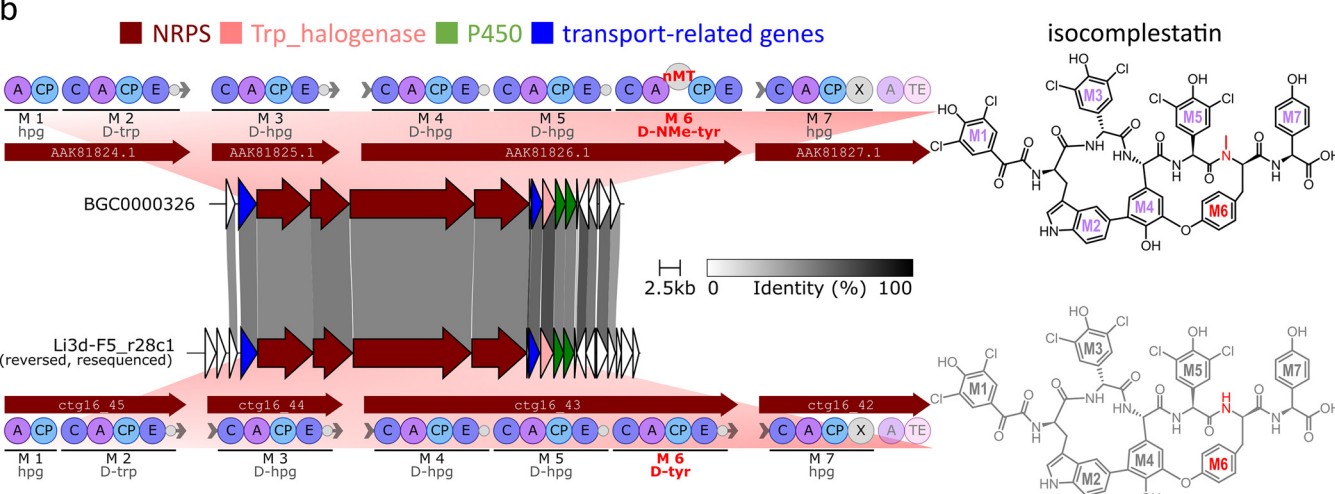

**FIG 8** GPA MF: (a) GPA MF extracted from Fig. 6b (green shaded area). (b) Through Clinker synteny analysis, we identified that one lichen BGC Li3d-F5_r28c1 is homologous to the reported GPA isocomplestatin BGC (left). Based on the domain organization comparison, the predicted compound of Li3d-F5_r28c1 should be a demethylated isocomplestatin (right). Compounds (right) were labeled to match the corresponding introducing modules on the assembly line (left).

**Conclusion.** The ever-decreasing cost of DNA sequencing and the continuous refinement of informatics tools for identifying and annotating BGCs within newly sequenced genomes means that genetics' first approaches are now an appealing means for the discovery of new bioactive natural products. As part of this search, it is essential that a variety of environmental niches are explored for the capacity to yield new bacterial strains and the natural products they produce. By coupling a cost-effective whole-genome sequencing workflow with a suite of informatics tools, we demonstrate that New Zealand's many lichen species are home to unique and rich populations of actinomycetes, and that these species are likely to be a fruitful avenue for future studies aimed at isolating and characterizing new compounds with potential medical relevance.

## MATERIALS AND METHODS

**General experimental procedures.** Lichens were sampled from different New Zealand locations (Fig. 1a; Data Set S1). Approximately 200 mg of each lichen sample was ground in a 1.5 mL microfuge tube using a micro pestle and suspended in 0.5 mL sterile 20% glycerol. Lichen samples were stored at −80℃. For isolation of actinomycetes, the lichen stocks were thawed and left to settle for 10 min on the bench. The resulting supernatants with 10-fold serial dilution were then plated on AIA plates (Sigma Aldrich, 17117-500G) and incubated at 30℃ for 14 days. The AIA plates were supplemented with nystatin (50 μg/mL) to inhibit fungi and nalidixic acid (50 μg/mL) to inhibit Gram-negative bacteria. Colonies with actinomycetes-like morphologies were restreaked onto fresh AIA plates. Following 14 days of incubation at 30℃, pure cultures/

spores were prepared and stored at −80°C in 30% glycerol as bacteria freezer stocks. Total DNA isolation was performed according to the salting-out protocol (40). Genomic DNA were stored in 50 $\mu$L of 5 mM Tris-HCL (pH 7.5, sterilized) in 96-well plates at 4°C. For each genomic DNA in this study, an in-house low-cost NGS Library was prepared using tagmentation enzyme (modified Tn5) that was recombinantly expressed and purified in-house as described in the protocol of Hennig (41). Each strain was given a unique forward and reverse barcode combination during PCR amplification of libraries. Following amplification, libraries were pooled and size selected using a 1% agarose gel. The final pooled library preparation had an average insert size of 500 bp as assessed using an Agilent 2100 bioanalyser. Sequencing was performed by GENEWIZ (Suzhou, China) using the HiSeq4000 platform.

**Genome assembly and annotation.** Raw reads were preprocessed using Trimmomatic (42) for quality trimming and adapter removal. The resulting filtered Illumina PE reads were then assembled using SPAdes 3.12.0 (43). Genome assembly quality was examined using CheckM (10) (Fig. S1a and b).

**Phylogenetic analysis and MASH distance calculation.** Taxonomic classification at the whole-genome level was assigned to each isolate using the identify-assign-classify workflow of GTDB-Tk v1.4.0 (11), and the taxonomic classification results were provided in Data Set S1. MASH (version 2.2.2) (12) was used for genome-wide ANI calculation for all-pairs comparisons between genomes. The ANI matrix generated using MASH was also used to cluster genomes. For dereplication, hierarchical clustering with average linkage and a distance cut-off of 0.5% identity was used. In order to cluster the genomes for which a species could not be assigned, hierarchical clustering with average linkage and a distance cut-off 4% identity was used. All trees (rectangular and circular) were produced *via* GTDB-Tk classify_wf, visualized using iTOL (44) (Branch lengths: Ignore).

**Cheminformatics analysis.** All entries in JSON format from MIBiG (41) (Version 2.0) were downloaded on May 27, 2021. MIBiG accession, Compounds (in SMILES string format), and Species (taxid) information were extracted from JSON files. Rdkit (v2020.09.1.0) (27) was used for molecular similarity network construction. Morgan Fingerprints of MIBiG compounds were generated with a radius of 2. Tanimoto similarity score was pairwise calculated between compounds, and an edge between compounds was created when the threshold of 0.5 was reached. Taxid from each MIBiG accession was used to fetch taxonomic information when queried NCBI Taxonomy (45). Compounds were marked by taxonomy as indicated in the MIBiG molecular similarity network (Fig. 6b).

**antiSMASH analysis.** Secondary metabolites were identified from assembled genomes using the standalone antiSMASH 5.1.0 (14), using prodigal as the genefinding-tool, and only processed sequences larger than 5,000 bp. A whole-genome HMMer analysis was also run for each genome. Identified clusters were compared against known reference BGCs from the MIBiG database alongside. An in-house python script was used to extract the following general information: "region"-"contig_edge" and "region"-"product" from output JSON files for downstream analysis. KnownClusterBlast is a module implemented in antiSMASH used for comparing each identified cluster against known MIBiG reference BGCs, with an output of MIBiG reference BGCs hits for each lichen query BGC. The similarity scores are sorted based on hits and synteny, a ranking system described by Medema et al. (28). An in-house python script was used to regenerate to the highest similarity using the following equation:

$$\frac{Number\ of\ hits\ in\ query\ region}{Number\ of\ query\ genes}$$

**Cblaster and Clinker.** Briefly, a local database was created from 4,309 actinobacteria assemblies using cblaster makedb module (18). A total of 8,541 lichen BGCs was then locally searched against the newly created local database. A binary table for each BGC containing the absence/presence of query genes in each homologous region from the local database was generated. The similarity is defined using the following equation, an equation adapted from MiBIG similarity score calculation:

$$\frac{Number\ of\ hits\ in\ query\ region}{Number\ of\ query\ genes}$$

The highest similarity is then reported/plotted in Fig. 2c.

BGC comparisons and visualizations were performed using Clinker with default settings (23).

**Gene cluster family analysis.** BGCs identified from antiSMASH 5.1.0 (14) and/or downloaded from MIBiG database (16) were first mapped back to the prebuilt GCF model calculated at clustering threshold (900) using BiG-SLiCE (version 1.1.0) (22) query mode. In-house python scripts were used to extract chemical class, GCF membership and distance values from SQLite3 database raw output file (Data Set S2). Identified BGCs were also clustered along with MIBiG BGCs (version 2.1) using BiG-SCAPE 1.1.0 (46) by applying cut-off = 0.6.

**Data availability.** The 322 lichen-associated actinomycetes read sets have been deposited to the NCBI Sequence Read Archive (SRA) under BioProject Accession PRJNA904503. The assembled contigs and antiSMASH output files (in json format) are available for download from https://doi.org/10.5281/zenodo.7349056. In-house python scripts for parsing antiSMASH results and BiG-SLiCE result can be accessed at https://github.com/phou0402/NZ_lichen.

## SUPPLEMENTAL MATERIAL

Supplemental material is available online only.

**FIG S1**, TIF file, 2.7 MB.

**FIG S2**, TIF file, 2.8 MB.

**FIG S3**, TIF file, 0.8 MB.
**FIG S4**, TIF file, 0.5 MB.
**FIG S5**, TIF file, 1.9 MB.
**FIG S6**, TIF file, 1.8 MB.
**DATA SET S1**, XLSX file, 0.05 MB.
**DATA SET S2**, XLSX file, 0.4 MB.
**DATA SET S3**, XLSX file, 0.02 MB.

## ACKNOWLEDGMENTS

We thank Liwei Liu for strain isolation. This work was supported by the Health Research Council of New Zealand (contract 16/172), the Royal Society of New Zealand Te Apārangi (contract RDF-VUW1601), and the Ministry for Business Innovation and Employment (contracts RTVU1908 and UOAX2010). The computations were performed on Rāpoi, the high-performance computing facility of Victoria University of Wellington.

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
