## [Reviewer comments · mSystems]

A genomic survey of the natural product biosynthetic potential of actinomycetes isolated from New Zealand lichens.

Peng Hou, Vincent Nowak, Chanel Taylor, Mark Calcott, Allison Knight, and Jeremy Owen

Corresponding Author(s): Jeremy Owen, Victoria University of Wellington

Review Timeline:

Submission Date:	October 27, 2022
Editorial Decision:	November 18, 2022
Revision Received:	December 6, 2022
Accepted:	December 15, 2022

Editor: Matthew Traxler

Reviewer(s): The reviewers have opted to remain anonymous.

Transaction Report:

DOI: <https://doi.org/10.1128/msystems.01030-22>

November 18, 2022

Dr. Jeremy George Owen
Victoria University of Wellington
School of Biological Sciences
TTR 410, Kelburn Parade
Kelburn
Wellington 6140
New Zealand

Re: mSystems01030-22 (A genomic survey of the natural product biosynthetic potential of actinomycetes isolated from New Zealand lichens.)

Dear Dr. Jeremy George Owen:

Thank you for submitting your manuscript to mSystems. I have reviewed your paper and I am pleased to inform you that, in principle, we expect to accept it for publication in mSystems. However, acceptance will not be final until you have adequately addressed the reviewer comments.

Preparing Revision Guidelines

Sincerely,

Matthew Traxler

Editor, mSystems

Journals Department
Reviewer comments:

Reviewer #3 (Comments for the Author):

Major Concerns:

Fig. 2: The outer stacked bar plot (3) is potentially problematic since genomes with high numbers of contigs will result in antiSMASH reporting an inflated number of BCGs. This is because a highly fragmented genome causes single BCGs to be detected as fragments on multiple contigs (as I think the authors are aware). So, I ask the authors to include a new ring on this diagram that displays the contig number. It could very easily be a heat map, like (or in place of) ring (2). Including this information is important to allow readers to see that strains that with high numbers of BCGs (like 40 or more) probably appear so due to high contig numbers, not because they are particularly enriched for BCGs.

Please provide a Data Availability paragraph. It should include relevant accession numbers, or a link where all genomic data can be accessed. Additionally, in-house scripts should be made available along with code documentation at a site such as Github. Appropriate links should be included here.

Minor Concerns:

P9 L171: "(7.59&:" should read "(7.59%:")

P12 L228: It's not really clear how hserlactone might translate into a unique defensive strategy since it refers to a quorum sensing molecule.

P26 L572: Figure 7 - CDA MF, please provide names of amino acids abbreviated in the figure.

Reviewer #3 (Comments for the Author):

Major Concerns:

Fig. 2: The outer stacked bar plot (3) is potentially problematic since genomes with high numbers of contigs will result in antiSMASH reporting an inflated number of BCGs. This is because a highly fragmented genome causes single BCGs to be detected as fragments on multiple contigs (as I think the authors are aware). So, I ask the authors to include a new ring on this diagram that displays the contig number. It could very easily be a heat map, like (or in place of) ring (2). Including this information is important to allow readers to see that strains that with high numbers of BCGs (like 40 or more) probably appear so due to high contig numbers, not because they are particularly enriched for BCGs.

Response: The reviewer raised a very pertinent point regarding the artifact of biosynthetic rich isolates. We have replaced the original Fig 2a (2) avg.len heatmap with a new (2) number of contigs heatmap.

And we have updated the main text and legend accordingly:

- lines 114-115 in the revised manuscript: *In total, we identified 8541 BGCs from 28,531 contigs (>5000bp, Figure 2a (2)). The average length of identified BGCs was 31.9 kb (Data Set S2).*
- lines 570-573 in the revised manuscript: *Additional information: (2) number of contigs longer than 5000 bp and (3) chemical classes were provided. Note that some of the biosynthetic 'rich' isolates were the*

artifact of the fragmented assembly. Assemblies that contain contigs only shorter than 5000bp were highlighted in grey.

We appreciate the reviewer's suggestion; we hope by making this change can convey a clearer message to the readers.

Please provide a Data Availability paragraph. It should include relevant accession numbers, or a link where all genomic data can be accessed. Additionally, in-house scripts should be made available along with code documentation at a site such as Github. Appropriate links should be included here.

Response: We have provided a Data Availability paragraph in the revised manuscript (lines 408-414):
Data availability

The 322 lichen-associated actinomycetes read sets have been deposited to the NCBI Sequence Read Archive (SRA) under BioProject Accession: PRJNA904503. The assembled contigs and antiSMASH output files (in json format) are available for downloading from <https://doi.org/10.5281/zenodo.7349056>. In-house python scripts for parsing antiSMASH results and BiG-SLiCE result can be accessed at https://github.com/phou0402/NZ_lichen.

Minor Concerns:

P9 L171: "(7.59&:" should read "(7.59%:")

Response: spelling error corrected (line 171 in the revised manuscript)

P12 L228: It's not really clear how hserlactone might translate into a unique defensive strategy since it refers to a quorum sensing molecule.

Response: We apologise for the confusion caused. We removed this sentence.

P26 L572: Figure 7 - CDA MF, please provide names of amino acids abbreviated in the figure.

Response: We have updated Fig 7d.

	1	2*	3	4	5	6		
BGC0000439 taromycin	6-Chloro-Tryptophan	D-Aspartic Acid	Aspartic acid	Threonine	Glycine	Ornithine		observed
BGC0000336 daptomycin	Tryptophan	D-Asparagine	Aspartic acid	Threonine	Glycine	Ornithine		
BGC0000291 A54145	Tryptophan	D- Glutamic Acid	3-hydroxyasparagine	Threonine	Sarcosine	Alanine		
Li4c-G7_r1c4	Alanine	D-X	3-hydroxyasparagine	Threonine	Sarcosine	Alanine		predicted
	7	8	9*	10*	11	12	13	
BGC0000439 taromycin	Aspartic Acid	D-Alanine	Aspartic Acid	Glycine	D-Alanine	3-methyl-glutamic acid	4-Chloro-kynurenine	observed
BGC0000336 daptomycin	Aspartic Acid	D-Alanine	Aspartic Acid	Glycine	D-Serine	3-methyl-glutamic acid	kynurenine	
BGC0000291 A54145	Aspartic Acid	D-Lysine	3-O-methyl-aspartic acid	Glycine	D-Asparagine	3-methyl-glutamic acid	Isoleucine/Valine	
Li4c-G7_r1c4	Aspartic Acid	D-Lysine	Aspartic Acid	Glycine	D-Asparagine	Glutamic acid	Proline	predicted

December 15, 2022

Dr. Jeremy George Owen
Victoria University of Wellington
School of Biological Sciences
TTR 410, Kelburn Parade
Kelburn
Wellington 6140
New Zealand

Re: mSystems01030-22R1 (A genomic survey of the natural product biosynthetic potential of actinomycetes isolated from New Zealand lichens.)

Dear Dr. Jeremy George Owen:

Your manuscript has been accepted, and I am forwarding it to the ASM Journals Department for publication. For your reference, ASM Journals' address is given below. Before it can be scheduled for publication, your manuscript will be checked by the mSystems production staff to make sure that all elements meet the technical requirements for publication. They will contact you if anything needs to be revised before copyediting and production can begin. Otherwise, you will be notified when your proofs are ready to be viewed.

Publication Fees:

If you would like to submit a potential Featured Image, please email a file and a short legend to msystems@asmusa.org. Please note that we can only consider images that (i) the authors created or own and (ii) have not been previously published. By submitting, you agree that the image can be used under the same terms as the published article. File requirements: square dimensions (4" x 4"), 300 dpi resolution, RGB colorspace, TIF file format.

We recognize that the video files can become quite large, and so to avoid quality loss ASM suggests sending the video file via <https://www.wetransfer.com/>. When you have a final version of the video and the still ready to share, please send it to mSystems staff at msystems@asmusa.org.

Thank you for submitting your paper to mSystems.

Sincerely,

Matthew Traxler
Editor, mSystems

Supplemental Material: Accept
Supplemental Material: Accept
Supplemental Material: Accept
Supplemental Material: Accept
Supplemental Material: Accept
Supplemental Material: Accept
Supplemental Material: Accept
Supplemental Material: Accept
Supplemental Material: Accept